# First Morphological and Molecular Identification of *Demodex injai* in Golden Jackal (*Canis aureus* Linnaeus, 1758) in Romania

**DOI:** 10.3390/pathogens12030412

**Published:** 2023-03-05

**Authors:** Sorin Morariu, Florica Morariu, Ana-Maria Marin, Maria Monica Florina Moraru, Dan-Cornel Popovici, Mirela Imre, Violeta Igna, Narcisa Mederle

**Affiliations:** 1Faculty of Veterinary Medicine, University of Life Sciences “King Michael I” from Timisoara, 300645 Timisoara, Romania; 2Faculty of Bioengineering of Animal Resources, University of Life Sciences “King Michael I” from Timisoara, 300645 Timisoara, Romania; 3Forestry Faculty, University Transilvania Brasov, 500036 Brașov, Romania

**Keywords:** golden jackal, PCR, *Demodex injai*, Romania

## Abstract

Demodicosis is one of the most important external parasitic diseases found in carnivores. Three species of the *Demodex* mite inhabit the skin of dogs and related species, *D. canis* being the most prevalent. This paper describes the first case of infestation with *D. injai* in a golden jackal in Romania. An emaciated golden jackal female body found in Timiș County, western Romania, was examined at Parasitology Department of Faculty of Veterinary Medicine, Timișoara. The gross lesions were present on different regions of the body: feet, tail, axillary and inguinal areas, and skin folds as well, consisting of erythema, extensive severe alopecia with lichenification, seborrhea, and scaling. In order to establish diagnosis, microscopic examination of skin scrapes, trichogram (hair plucking), acetate tape test (impression), fungal culture, and PCR were performed. Both microscopic measurements and PCR analysis have confirmed the presence of *D. injai*.

## 1. Introduction

The golden jackal (*Canis aureus* Linnaeus, 1758) is a wild carnivore belonging to the Canidae family, which, in recent years, because of its ecological plasticity, has expanded its European range from its south to the center [1]. It is currently considered an invasive species, which managed to populate in a short time in territories in Turkey, Greece, Bulgaria, Serbia, and Hungary, being a common species in Romania as well, where it is frequently found from Dobrogea (eastern Romania) to Banat (western Romania) [2]. At the same time, the species found in Europe and Asia are genetically different from their African relative [3].

The dietary and behavioral peculiarities of the golden jackal make it a recognized reservoir for various pathogens, including parasites, which it can spread to new areas [2,4,5]. Although studies on the presence of internal parasites in this species in Europe are sufficient [6,7,8,9,10,11,12], those regarding external parasites and the skin conditions they produce are parsimonious. Most reports refer to the presence of various species of ixodid ticks [13,14,15,16]; only a few from all over the world refer to dermatitis produced by species of the genus *Demodex* (Acariformes: *Demodecidae*) [17,18].

In general, dermatitis refers to the various inflammations and/or irritations of the skin found in both humans and animals, some of which are zoonotic and which, in free-living wild animals, can induce severe manifestations, sometimes culminating in the death of the affected individual, since the management of such episodes is difficult to achieve [19,20]. 

Three genera of mites: *Sarcoptes*, *Notoedres*, and *Demodex* constitute the main cause of dermatitis in wild animals. The first two are represented by highly contagious mites and the genus *Demodex* is represented by mites that are considered (excessively) commensal, but which, when proliferating, can produce severe dermatitis, known as demodicosis [19,21,22].

Given the paucity of data on the species of *Demodex* that can be found in the golden jackal, this paper describes the first identification, both morphological and molecular, of *Demodex injai* in this wild carnivore species.

## 2. Materials and Methods

### 2.1. Case Report

#### 2.1.1. Clinical Examination

A corpse of a golden jackal female almost 2 years old, found in a silvo-steppe area in the vicinity of a small town in the south of Timiș County, was examined in the Parasitic Diseases clinic of FVM Timișoara. Unfortunately, the cause of death has not been established. The animal was severely emaciated and presented erythema and extensive severe alopecia accompanied by lichenification, seborrhea, and scaling. The lesions were located on different areas of the body: skin folds, feet, tail, and axillary and inguinal regions (Figure 1).

#### 2.1.2. Diagnosis

After clinical examination, some laboratory tests such as skin scrapes, microscopic examination, trichogram (hair plucking), acetate tape test (impression), and fungal culture were performed in order to establish the diagnosis.

Deep skin scrape is commonly used to identify mites such as *Demodex* spp., *Sarcoptes* spp., and *Notoedres* spp. This collecting technique involves obtaining material from the edge of the lesions, which is mixed with a clarifying liquid and homogenized on a microscope slide, then covered with a coverslip and examined with the 10× microscope objective.

A trichogram has the same objectives as the deep scrape, to which is also added to the identification of the presence of dermatophytes. A small clump of hairs from the edge of the lesions is plucked after which it is placed on a microscope slide, followed by other steps identical to those of the skin scrape.

Acetate tape testing is convenient and reliable. A clear adhesive tape is pressed onto the surface of the skin to collect keratinocytes, any superficial microbes, and other ectoparasites such as *Cheyletiella*, a non-burrowing mite.

Fungal culture was performed by inoculating the hair and crust samples into the dermatophyte test medium (DTM) plate. The plate was incubated at 27 °C for 12 days.

Measurements were also performed in order to establish the specific sizes of the different developmental stages of mites (Table 1). They were made for 16 adults (11 males and 5 females), 5 nymphs, 5 larvae, and 8 eggs (Figure 2).

#### 2.1.3. DNA Extraction and Molecular Analysis

The PCR reaction was carried out according to the technique described by Frank et al. 2013 [23] and was also used to identify *D. cati* in the same location [24]. The amplification was carried out by classical PCR and was based on the amplification of a sequence of the ~330 bp (fragment of the 16S rDNA) for *Demodex* spp., modified for the requirements of the mixture.

According to the protocol, both 5’ACTGTGCTAAGGTAGCGAAGTCA3’ forward primer and 5’TCAAAAGCCAACATCGAG3’ reverse primer [23] were used. They are considered highly conservative for several *Demodex* species, mainly *D. gatoi*, *D. caprae*, *D. brevis*, *D. folliculorum*, *D. canis*, and *D. injai* [23,25]. The reaction used a Master Mix MyTaqTM Red Mix (BIOLINE^®®^, Cincinnati, OH, USA) to achieve the results. The final volume of the PCR reaction was 25 μL, of which 12,5 μL MyTaqTM Red Mix (BIOLINE^®®^), 1 μL 1st primer, 1 μL 2nd primer (diluted to a concentration of 10 pmol/μL according to the protocol described by the manufacturer), and DNA extracted from the sample was analyzed, as well as ultrapure water. Amplification was accomplished with the My Cycler thermocycler (BioRad^®®^). This program included a stage of DNA denaturation at 95 °C for 90 sec, followed by 35 cycles of 55 °C for 30 s, 68 °C for 120 s, and 94 °C for 30 s, one of 55 °C for 30 s, and a final cycle 68 °C for 5 min. The analysis and control of amplicons was performed by horizontal electrophoresis in a submerse system of electrophoresis in 1.5% agarose gel, with the addition of flour dye MidoriGreen (Nippon Genetics^®®^ Europe Gmbh, Düren, Germany) to a voltage of 120 V and 90 mA for 60 min. A 100 bp size Ladder DNA marker was used in the first well of the gel. The gel image with the migrated DNA fragments was captured using an UV photodocumentation system (UVP^®®^).

PCR products were sequenced at Macrogen Europe^®®^ Company (Amsterdam, The Netherlands) and compared with those available in the GenBank database, using BLAST alignment (Appendix A).

## 3. Results

The laboratory tests revealed the presence of some cocci and rods, which were associated with *Staphylococcus* spp., respectively *Pseudomonas aeruginosa* and a few *Malassezia* spp. organisms. None of the usual ringworms grew on the DTM.

The scrapes proved positive for the presence of mites. Based on specific morphological characteristics (Table 1), the identified mites were classified as species of *Demodex injai* (Figure 2).

Thus, the total length of the adult mite was 238.56 µm and on segments, it was 25.53 µm for the gnathosoma, 63.05 µm for the podosoma, and 135.62 µm for the opisthosoma, respectively. The deutonymph measured 218.69 µm, the larval stage 137.89 µm, and the egg 65.11/25.64 µm.

## 4. Discussion

It is known that *D. injai* is the largest species of *Demodex* found in carnivores, especially in dogs. Basically, it is a prostigmatous mite with a vermiform appearance, which lives in the pilosebaceous follicles of the hosts and which, following intense proliferation, produces a chronic dermatological condition, sometimes generalized, rarely relapsing, called demodicosis. It usually results in folliculitis and/or furunculosis, accompanied by hyperseborrhea, erythema, hyperpigmentation, alopecia, comedones, and a characteristic odor of the affected areas [26,27,28].

In a study carried out in Poland between 2001 and 2008, *D. injai* was identified in two dog skin samples out of 39 examined and collected post-mortem [29]. The average mite length was 367 µm (309–411 µm) for the male and 339 µm (282–396 µm) for the female, three times higher that of *D. cornei* and almost double that of *D. canis*. Another study carried out more recently in 2020 by Chaudhary et al. [30], based on the morphometric analysis of 50 individuals of *D. injai*, revealed the following average dimensions of the parasite: gnathosoma—22.42 ± 0.60 µm, podosoma—74.94 ± 0.77 µm, opisthosoma—166.24 ± 2.55 µm, and total length—263.61 ± 2.83 µm. Samples were collected from 40 clinically manifested dogs in the Indian Veterinary Research Institute, Izatnagar, Bareilly, India. These sizes are smaller than those reported by Izdebska [29] in 2010 and close to those measured in our study.

Using the PCR technique, the result was identical, finding that the species involved was *D. injai*. Products of approximately 330 bp were obtained after PCR amplification. The rRNA sequences obtained from *D. injai* in the hind legs and axillary and inguinal areas were similar to those reported from morphologically alike parasites obtained from the skin. In the BLAST search, our sample was closely matched (96%) to the GenBank dog origin *D. injai* sequence (HE 817765) from Spain. 

This case represents the first identification of *D. injai* in the golden jackal in Romania. Although initially the diagnosis was based on microscopic examination, confirmation and exact identification of the mite was achieved by PCR. However, it seems that the dimensions of the same parasite are variable depending on the body area from which it is collected [31]. This can, in fact, explain the differences in size recorded in the studies mentioned above [29,30]. *D. injai* was more frequently identified in terriers, especially in those with compromised immune status, in which seborrheic dermatitis of the dorsal area of the trunk was described [32,33].

In the case of the golden jackal, the specialized literature is very poor regarding demodicosis. In Israel, a case of simultaneous parasitism with two species of mites (*Sarcoptes scabiei* and an unnamed species of *Demodex*) was described, manifested as a chronic diffuse but severe dermatitis and folliculitis [17]. Another study carried out in the Caucasus area identifies the presence of *D. canis* in five out of 150 examined jackals, with a prevalence of 3.33% [2]. Controversial information about the presence of *Demodex spp*. was provided from the examination of five golden jackal specimens in Bangladesh, without specifying the mite species, prevalence, and intensity of parasitism [18].

The etiopathogenetic mechanisms of *Demodex* infection are not yet very well known, but it seems that a dysfunction of CD4+ T lymphocytes triggers the disease. This dysfunction may be due to parasitic co-infections, endocrine diseases (hypothyroidism, hyperadrenocorticism, or diabetes mellitus), neoplasms, or allergies that cause immunosuppression [34]. Thus, the synergy between immune factors and other intrinsic and extrinsic stressors is determinant in breaking the balance between the parasite and the host [35]. 

## 5. Conclusions

Analyzing data obtained from the specialized literature, the present study reports *D. injai* infection in the golden jackal for the first time in Europe. The increasing density of these populations highlights their potential role in the transmission of *D. injai* to congeners or to other species of carnivores with which it comes into contact, together with other zoonotic parasites. Additionally, they contribute to the maintenance of natural disease foci.

## Figures and Tables

**Figure 1 pathogens-12-00412-f001:**
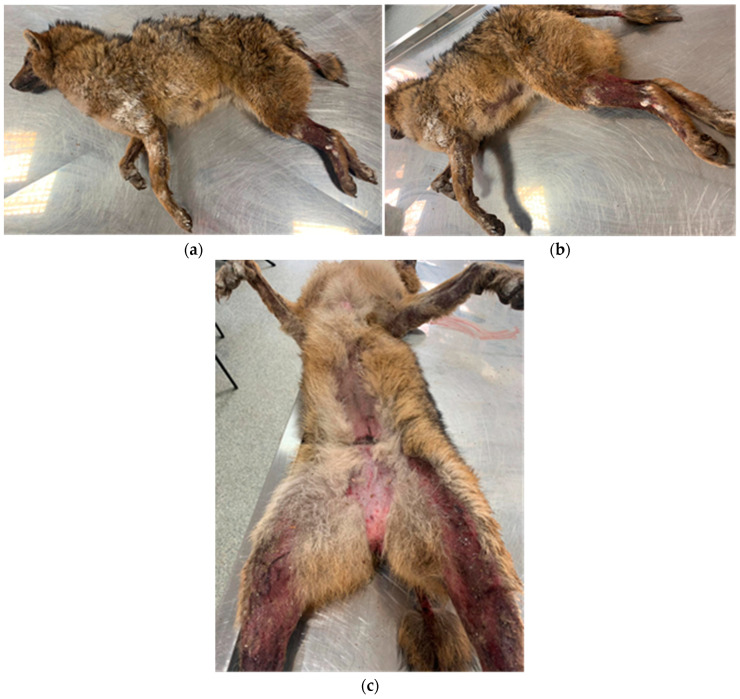
Extensive alopecic areas on fore- and hind limbs, on the ventral abdomen and inguinal, including the tail, with erythema, hyperpigmentation, lichenification, and some follicular papules and pustules; (**a**,**b**)—lateral view of the entire body; (**c**)—enlarged view of the inguinal area and the medial part of the hind limbs.

**Figure 2 pathogens-12-00412-f002:**
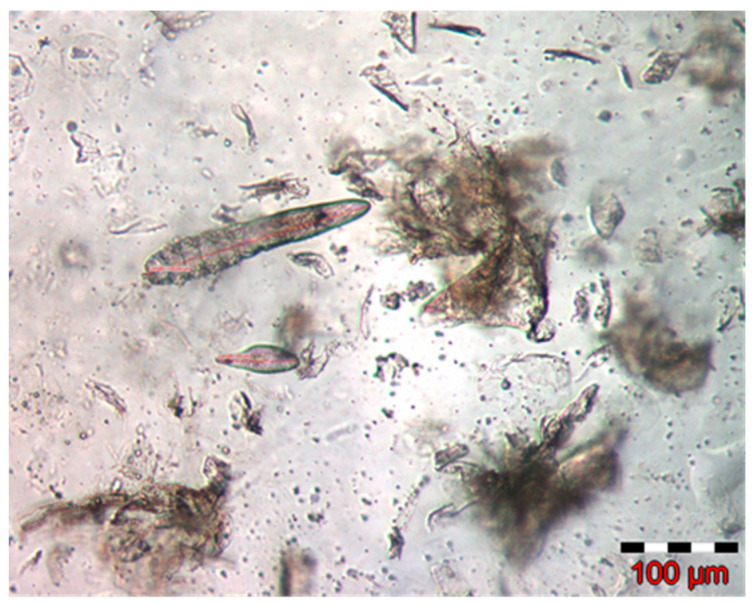
Nymph and egg of *Demodex injai*.

**Table 1 pathogens-12-00412-t001:** *Demodex* mites’ measurements according to their developmental stages.

Developmental Stage	Total Length µm (min.–max.)	Gnathosoma µm (min.–max.)	Podosoma µm (min.–max.)	Podosoma Width µm (min.–max.)	Opisthosoma µm (min.–max.)	Egg Width µm (min.–max.)
Adult (*n* = 16)	238.56 (203.86–249.19)	25.53 (18.71–27.95)	63.05 (56.31–77.20)	36.41 (33.17–40.19)	135.62 (109.37–157.10)	-
Nymph (*n* = 5)	218.69 (196.23–234.69)	23.72 (17.66–26.33)	61.49 (61.60–63.89)	38.65 (33.04–44.49)	134.45 (113.64–150.68)	-
Larva (*n* = 5)	137.89 (98.67–190.30)	-	-	29.54 (23.35–36.78)	-	-
Egg (*n* = 8)	65.11 (56.85–71.78)	-	-	-	-	25.64 (20.25–32.08)

## Data Availability

Not applicable.

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
