# Peer review of "First Morphological and Molecular Identification of Demodex injai in Golden Jackal (Canis aureus Linnaeus, 1758) in Romania"

_pathogens, 2023, doi:10.3390/pathogens12030412_

Round 1
Reviewer 1 Report
I would like to congratulate the authors and this very interesting and novel study, there are some remarks that must be addressed before this paper will be ready for publication, i hope the authors will find them helpful:
Line 31-34: Provide reference for this statement
Line 48: Replace because with since
Line 50: rephrase the whole statement due to grammar mistakes and incorrect choice of words, suggestion is "Three genera of mites... constitute the main cause of dermatitis for wild animals"
Line 51: Remove if
Figure 1 needs restyling by using small letters for the description of different segments and also by using transparent background for the letters, also by unifying their position of the pictures, this will add to the proper representation of the paper (Fig 1 is chaotic a bit in its current state)
Line 59: Did the authors establish the cause of death? Age? collections sit (Forest, urban, street etc..? These are relevant questions to the spread and transmission of the parasite.
Line 107-108: Provide primers in 5'-3' format
Line 156: Where was the study conducted? hosts? year? were these demodex found free living? the current statement is vague and short
Line 165-166: Percentage of similarity and the origin of the reference sequences are ought to be provided here in the text, rather than the supplementary files in the form of text
Line 189: It is recommended to remove this phrase, the authors cannot make this statement if they did not conduct any immunological or endocrinal tests on the carcass in addition to the fact that the cause of death is unknown (Not mentioned earlier)
Why was the sequence not deposited into the GenBank? it is very important to deposit it and add the accession number to the materials and methods section since this is a novel case and will help future studies
Author Response
Please, see the attachment.

Reviewer 2 Report
Methodology
Lines 61-64: Please describe more details of the animal body that was recovered and add more details of ethical documents to allow access to these samples.
Line 82: Demodex spp., Sarcoptes spp. and Notoedres spp. or identify genera of mites like…
Results:
Line 129: add dot after species abbreviation: spp.
Author Response
Please, see the attachment.
